# Plants Used in Mexican Traditional Medicine for the Management of Urolithiasis: A Review of Preclinical Evidence, Bioactive Compounds, and Molecular Mechanisms

**DOI:** 10.3390/molecules27062008

**Published:** 2022-03-21

**Authors:** Delia Sansores-España, Alfredo Geovanny Pech-Aguilar, Karol Guadalupe Cua-Pech, Isabel Medina-Vera, Martha Guevara-Cruz, Ana Ligia Gutiérrez-Solis, Juan G. Reyes-García, Azalia Avila-Nava

**Affiliations:** 1Sección de estudios de Posgrado e Investigación, Escuela Superior de Medicina, Instituto Politécnico Nacional, Ciudad de México 11340, Mexico; deliasansores@hotmail.com (D.S.-E.); juangreyesgarcia@gmail.com (J.G.R.-G.); 2Hospital Regional de Alta Especialidad de la Península de Yucatán, Yucatán, Mérida 97130, Mexico; alfredopsp97@gmail.com (A.G.P.-A.); kcuapech@gmail.com (K.G.C.-P.); ganaligia@gmail.com (A.L.G.-S.); 3Escuela de Salud, Universidad Modelo Campus Mérida, Yucatán, Mérida 97305, Mexico; 4Becario de la Dirección General de Calidad y Educación en Salud (DGCES), Secretaría de Salud, Ciudad de México 11410, Mexico; 5Departamento de Metodología de la Investigación, Instituto Nacional de Pediatría, Ciudad de México 04530, Mexico; isabelj.medinav@gmail.com; 6Tecnologico de Monterrey, Escuela de Medicina y Ciencias de la Salud, Ciudad de México 14380, Mexico; marthaguevara8@yahoo.com.mx; 7Departamento de Fisiología de la Nutrición, Instituto Nacional de Ciencias Médicas y Nutrición Salvador Zubirán, Ciudad de México 14080, Mexico

**Keywords:** urinary stones, Mexico, traditional plants, bioactive compounds

## Abstract

Urolithiasis (UL) involves the formation of stones in different parts of the urinary tract. UL is a health problem, and its prevalence has increased considerably in developing countries. Several regions use plants in traditional medicine as an alternative in the treatment or prevention of UL. Mexico has known about the role of traditional medicine in the management of urinary stones. Mexican traditional medicine uses plants such as *Argemone mexicana* L., *Berberis trifoliata Hartw. ex Lindl.*, *Costus mexicanus Liebm*, *Chenopodium album* L., *Ammi visnaga* (L.) *Lam.*, *Eysenhardtia polystachya* (*Ortega*) *Sarg.*, *Selaginella lepidophylla* (*Hook. & Grev.*) *Spring*, and *Taraxacum officinale* L. These plants contain different bioactive compounds, including polyphenols, flavonoids, phytosterols, saponins, furanochromones, alkaloids, and terpenoids, which could be effective in preventing the process of stone formation. Evidence suggests that their beneficial effects might be associated with litholytic, antispasmodic, and diuretic activities, as well as an inhibitory effect on crystallization, nucleation, and aggregation of crystals. The molecular mechanisms involving these effects could be related to antioxidant, anti-inflammatory, and antimicrobial properties. Thus, the review aims to summarize the preclinical evidence, bioactive compounds, and molecular mechanisms of the plants used in Mexican traditional medicine for the management of UL.

## 1. Introduction

Urolithiasis (UL) is the presence of stones in the kidney, ureters, bladder, and/or urethra. It is a multifactorial phenomenon where there are one or more alterations in the urine composition that promote crystallization [1,2]. The formation of urinary stones is a result of the supersaturation of urine and the subsequent formation of crystalline materials. The formation of urinary stones involves a sequence of events, including saturation, supersaturation, nucleation, crystal growth, crystal aggregation, crystal retention, and, finally, stone formation [3,4]. During urinary supersaturation and crystallization, there is intrarenal crystal precipitation, which is mainly caused by inherited or acquired diseases associated with renal function impairment [4,5]. During UL, many factors contribute to urinary stone formation, including alteration in urinary pH and metabolic alterations such as hypercalciuria, hyperuricosuria, hyperoxaluria, hypocitraturia, hypomagnesuria, and hypercystinuria [2,3]. These factors are related because the urinary pH promotes excessive concentrations of many compounds such as calcium, oxalates, phosphates, uric acids, urates, struvite, amino acids, and purines, which, subsequently, results in supersaturation and crystallization [4]. The chemical composition of urinary stones depends on the biochemical abnormalities in urine composition. Calcium oxalate (CaOx) is the most common component of urinary stones; it represents up to 80% of analyzed stones [2]. Other types of stones that are less frequent are calcium phosphate, struvite (5–15%), uric acid (5–10%), and cystine stones that correspond to 1–3% of stone types [4]. 

The prevalence of UL has increased from 3.2% in 1976 to 10.1% in 2016; thus, it has become a public health issue. There is evidence about the increasing incidence of stones to 7–13% in North America, 5–9% in Europe, and 1–5% in Asia [5,6]. In Mexico, there are few epidemiological studies about this pathology. One of these studies by the Mexican Institute of Social Security reported an average prevalence of 2.4/10,000 inhabitants. However, recently, it has been reported that the prevalence is 550/10,000 [7]. Approximately 13 of every 1000 hospital discharges globally are due to UL. UL is a frequent cause of hospital admission in the emergency services and can trigger various complications, resulting in an economic burden for the health system [5]. There is variation in its prevalence according to geographical characteristics, socioeconomic status, and gender. Currently, the prevalence of UL in men and women is 13% and 7%, respectively [8]. Many studies have shown that the increased prevalence and variation in prevalence are associated with environmental and metabolic risk factors, such as diet, physical inactivity, metabolic disorders, urine composition and volume, urinary tract infections, and genetic predisposition [8,9]. 

Even though half of the cases of symptomatic kidney stone events could be prevented by the modification of risk factors [10], medical management and pharmacological therapies (allopurinol, citrate, cystone, and thiazide diuretics), unfortunately, are not effective in all cases. On the other hand, surgical treatment is not fully effective either, as it could promote long-term renal damage, hypertension, and kidney stones [11,12]. Moreover, these available synthetic drugs have many adverse effects, and in some areas in the world, it is difficult to access these conventional drugs. Therefore, it is necessary to use alternative treatments or the development of new antilithiatic drugs with fewer side effects [13]. 

Several regions in traditional medicine include the use of plants as an alternative in the treatment or prevention of UL [14]. In particular, Mexico has had a great knowledge of traditional medicine since ancient times, used by the Aztecs before the arrival of the Spanish. In Mexico, there are more than 23,400 vegetable plants and 5000 species that are used for medicinal purposes. More than 90% of Mexicans use remedies based on medicinal plants as empirical treatments of various diseases due to their effectiveness, tradition, and low cost [15,16]. However, a limitation in the use of medicinal plants is the lack of information about the recognition of the bioactive compounds and their effects on UL. Moreover, the action mechanism of many phytotherapeutic agents is unknown. Thus, the present review aimed to summarize the preclinical evidence, bioactive compounds, and molecular mechanisms of the plants used in Mexican traditional medicine in the management of UL. Thus, this information could provide information for the design of clinical trials to test the efficacy and safety of this type of treatment.

## 2. Molecular Mechanisms Involved in Urinary Stone Formation

It is important to understand the molecular mechanisms of urinary stone formation to treat patients appropriately, as this will reduce the morbidity and healthcare costs associated with UL [17]. The multifaceted process of UL involves a sequence of events in the formation of urinary stones. Among the molecular mechanisms involved in the formation of urinary stones are inflammation and oxidative damage caused by oxidative stress (OS) (Figure 1). During OS, there is an increase in the levels of reactive oxygen species (ROS) and a decreased antioxidant system [18]. It is known that inflammation and OS facilitate crystal adhesion to renal tubular cells and crystal formation, promoting kidney stone development [19]. Excessive levels of ROS are important for cellular signaling, including the activation of pro-inflammatory genes expression and impaired endothelial function [20]. Additionally, an excessive level of ROS causes the generation of isoprostanes and prostaglandins, which increase the inflammatory environment and can induce apoptosis [18]. 

It has been shown that the exposure of renal epithelial cells to oxalate leads to alterations in the normal activities of the renal epithelium, causing changes such as impaired mitochondrial function and increased ROS level [21]. Under normal physiologic conditions, the balance with ROS is maintained by antioxidant systems, including enzymes, such as superoxide dismutase (SOD), glutathione peroxidase (GPx), catalase, thioredoxin, peroxiredoxin, and glutathione transferase [22,23]. However, during UL, a reduction in this antioxidant system promotes a disequilibrium in redox status. A high level of ROS promotes the oxidation of renal cell membrane proteins and lipids, causing biochemical alterations and induced inflammatory processes. Some studies demonstrated that oxalate and CaOx crystals might cause damage in renal tubular cells. This effect could be due to an increase in the release of lysosomal enzymes and cytosolic enzymes [1,24,25].

Additionally, excessive ROS levels trigger the activation of the nuclear transcription factor κB (NF-κB). This factor induces the expression of pro-inflammatory genes such as kidney injury molecule 1, the proliferation of cell nuclear antigens, and the homing of cell adhesion molecules (CD44). These chemokines consequently enhance the recruitment of various immune cells, including monocytes, macrophages, neutrophils, dendritic cells, and T-cells, which are associated with injury in the kidneys [26,27]. Several studies have demonstrated that macrophage-derived exosomes following calcium oxalate monohydrate (COM) exposure are involved in urinary stone pathogenesis [4,28,29]. Macrophage accumulation and macrophage-related inflammation, the main immune response alterations observed in UL, have been widely reported to play a crucial role in renal CaOx crystal formation [28]. The recruited macrophages could promote the development of CaOx crystals via the interaction of CD44 with fibronectin, which is upregulated in renal tubular cells induced by crystals [26]. Then, macrophages secrete various mediators via several classical molecules to generate a pro-inflammatory state such as macrophage inhibitory protein-1, monocyte chemoattractant protein-1, and interleukin-8 [29]. These biochemical and structural alterations caused by ROS and inflammation result in the loss of membrane integrity promote fibrosis and collagen formation and facilitate CaOx adhesion and retention. There is increasing evidence demonstrating that the tissue injury induced by OS enhancing the retention of the causative crystals inside renal tubules and/or kidney interstitium (parenchyma) is one of the crucial steps for kidney stones [21]. This dysfunction is related to not only renal tubular injury but also the impaired immune response and inflammation by decreasing the mitochondrial functions in the CaOx stone formers, leading to the decline in crystal elimination that further enhances tissue inflammation [30,31,32]. Progressive tissue inflammation, together with the supersaturation of calcium phosphate, induces Randall’s plaque formation, the main mechanism involved in the stone formation process, urine exposure, and CaOx deposition on the exposed plaque surface, and may ultimately be the trigger for urinary stone development or growth [33]. Due to the high prevalence of UL and the role of OS and inflammation in urinary stone formation, it is important to know in detail the alternatives for treatment or prevention of this pathology.

## 3. Plants Used in Mexican Traditional Medicine against Urolithiasis

In Mexico, the use of medicinal plants such *as Argemone mexicana* L., *Berberis trifoliata Hartw. ex Lindl.*, *Costus mexicanus Liebm*, *Chenopodium album* L., *Ammi visnaga* (L.) *Lam.*, *Eysenhardtia polystachya* (*Ortega*) *Sarg.*, *Selaginella lepidophylla* (*Hook. & Grev.*) *Spring*, and *Taraxacum officinale* L. to treat UL has been reported (Appendix A). These types of plants contain bioactive compounds such as flavonoids, phytosterols, saponins, furanochromones, alkaloids, and terpenoids, which can have a positive effect on UL due to their antimicrobial, litholytic, antispasmodic, diuretic, antioxidant, and anti-inflammatory properties, which promote a reduction in urinary saturation by inhibiting crystallization, nucleation, and crystal aggregation [34] (Table 1).

### 3.1. Chicalote Amarillo (Argemone mexicana *L.*)

*A. mexicana* L., belonging to the *Papaveraceae* family, is considered an important medicinal plant. This plant is an exotic weed indigenous named “chicalote amarillo” and is distributed in South America but has a widespread distribution in many tropical and sub-tropical countries. In traditional medicine, it is useful in treating dropsy, jaundice, ophthalmia, scabies, and urinary stones [35]. An in vitro study showed that the methanolic extract of *A. mexicana* L. leaves promoted a significant inhibition of nucleation (72.3%) and aggregation (77.2%) compared to the cystone and drug (62.9 and 69.33%, respectively) [36]. In another study in an animal model of UL induced by ethylene glycol (EG), the oral administration of an extract from *A. mexicana* L. (400 and 750 mg/kg) for 13 days promoted regulation of the concentration of serum urea, nitrogen, and creatinine. Additionally, this extract protected from renal damage and increased diuresis activity, thereby, completely excreting CaOx crystals in the renal tubes [37]. 

This physiological effect is explained by the presence of many bioactive compounds in *A. mexicana L*. The anti-urolithiatic activity may be attributed to the presence of compounds such as alkaloids, terpenoids, and flavonoids; specifically, to its bioactive compound 3,4-dihydro3-hydroxy-7-(7-methyloctyl) naphthalen-2(1H) [38].

### 3.2. Palo Amarillo (Berberis trifoliata Hartw. ex Lindl.)

*Berberis trifoliata Hartw. ex Lindl*, belonging to the *Berberidaceae* family, has been used as a part of traditional medicine in northeastern Mexico, and it is known as “Palo amarillo”. This plant has wide medicinal applications, including laxative and anti-urolithiatic activity. Additionally, it prevents urinary stone formation. An in vitro study showed that methanolic extracts from this plant (1000 μg/mL) inhibited the nucleation of crystals of CaOx (93 ± 0.01%) [39]. In another study in an animal model of UL induced by zinc disc implantation in the urinary bladder, the anti-urolithiatic activity of the methanolic extract was demonstrated. This study showed that the oral administration of different concentrations of extract (50, 100, and 150 mg/kg) for 20 days decreased the formation and the weight of the dry stones removed from the urinary bladder compared with the control group. Furthermore, this study showed that intervention with methanolic extract of *B. trifoliata* promoted a significant reduction in the crystal depositions in a model of UL induced by the implantation of a zinc disc in the bladder [40]. 

These beneficial effects could be associated with the presence of anthocyanins and alkaloid compounds, mainly berberine. This compound has been identified as a component responsible for potential therapeutic effects, including an anti-urolithiatic effect [40,41]. 

### 3.3. Planta de la Insulina (Costus Mexicanus Liebm)

*Costus mexicanus Liebm* belongs to the *Costaceae* family, it is commonly known as “planta de la insulina”, fiery costus, step ladder, or spiral flag. This plant is used in traditional medicine as an infusion in the treatment of renal disorders [42,43]. An in vitro study showed that an aqueous extract of *Costus mexicanus* Liebm. has a beneficial effect on the nucleation and crystallization of CaOx crystals. The results showed that the increase in the concentration of these aqueous extracts (0.15% to 1.00%) decreased the mass of these crystals by 96.7%. Additionally, the aqueous extract inhibited the formation of COM crystals, one of the major components of urinary calcium stones [43].

On the other hand, a study showed the effects of aqueous and ethanolic extracts of *Costus mexicanus Liebm* and isolated compounds lupeol and stigmasterol in a model of hyperoxaluria induced by EG. The results showed that the oral administration of aqueous or methanolic extracts of the stem of *Costus mexicanus Liebm* (100 mg/kg) for 28 days significantly decreased the serum concentration of urea, uric acid, calcium, phosphate, and creatinine to a near normal condition. Moreover, the treatment with lupeol and stigmasterol reduced the deposition of calcium and oxalate in kidney tissue [44]. 

The effects observed by the treatment with this plant could be due to the presence of several bioactive compounds, among them lupeol and stigmasterol, which have been associated with antioxaluric and anticalciuric activities [44]. 

### 3.4. Cenizo (Chenopodium album *L.*)

*Chenopodium album* L. is a member of the family *Amaranthaceae,* and it is usually known as “Cenizo”. Its leaves are traditionally used to correct kidney diseases and urinary stones. An in vitro study showed the inhibitory effect of the aqueous extract of the leaves from this plant (500 and 1000 μg/mL) on the crystallization, nucleation, and aggregation of CaOx and brushite crystals [45]. Another study in an animal model of UL showed the beneficial effect of daily administration of methanolic or aqueous extracts of *Chenopodium album* L. leaf (400 mg/kg). The results showed that these extracts significantly reduced the plasma levels of calcium, phosphorus, urea, uric acid, creatinine, and the pH and oxalate volume in urine. Furthermore, this intervention caused a reduction in the deposition of the crystals and the amount of oxalate in renal tissue [46].

The beneficial effects of this plant could be associated with constituents such as flavonoids and saponins, which can act as antioxidants and anti-crystallization compounds [45,46].

### 3.5. Visnaga (Ammi visnaga (*L.*) Lam.)

*Ammi visnaga Lam* belongs to the family *Apiaceae;* its common name is “Visnaga”. Studies have focused on the diuretic activity of *A. visnaga*, which is effective in the treatment of UL [47,48]. The effect of the *A. visnaga* extract against UL has been studied in different models. An in vitro study in human urine supersaturated with CaOx showed that an extract of seeds of Visnaga (125 mg/kg) reduced the crystallization of CaOx (up to 24%). However, the extract of the whole plant did not show reductions [49]. Thus, results suggested that the beneficial effects may be due to the presence of specific bioactive compounds such as khellin and visnagin in a special part of the plant. A study in an animal model of hyperoxaluria showed that the oral administration of the aqueous extract of khella (125–500 mg/kg) for 14 days significantly reduced the incidence of CaOx crystal deposition in the histopathological examination of kidneys. In addition, this intervention increased the urinary pH and volume, as well the urinary citrate [50]. These effects may be related to changes in luminal pH, which inhibits citrate absorption and prevents the formation of CaOx. Another in vitro study showed that khellin extract (200 mg/mL) reduced the incidence of CaOx kidney deposition in renal epithelial cells exposed to oxalate (300 mM) or COM (133 mg/cm^2^) [51]. 

### 3.6. Palo Azul (Eysenhardtia polystachya (Ortega) Sarg.)

*Eysenhardtia polystachya (Ortega) Sarg*, a tree native from Mexico, belongs to the family *Fabaceae;* it is commonly known as “palo azul” or “palo dulce”. It is widely used in popular medicine for its antitussive, antispasmodic, febrifuge, anti-inflammatory, and antirheumatic properties. Additionally, it is used in renal disease due to its diuretic and antimicrobial activities [52,53]. Scientific evidence showed the beneficial effects of intervention with *E. polystachya* against urinary stones. A study showed the antilithiatic and diuretic activity of an extract of this plant in an animal model of UL induced by the implantation of a zinc disc into the urinary bladder. The extract (50–100 mg/kg) promoted a significant decrease in the weight of stones and, in 24 h, increased urine volume [52]. Another study, in a similar model, showed that the oral administration of an aqueous extract from *Eysenhardtia polystachya* (*Ortega*) *Sarg.* (500 and 750 mg/kg) after six hours in metabolic cages promoted an increased urinary flow, similar to the control group treated with pharmacological treatment (furosemide) [53]. 

The beneficial effects of *E. polystachya* have been attributed to its content of bioactive constituents as flavonoids, saponins, and tannins [53]; mainly by the presence of 7-hydroxy-2′,4′,5′-trimethoxyisoflavone and 7-hydroxy-4′-methoxyisoflavone, which can inhibit the formation and growth of crystals, reducing the occurrence of UL [54]. 

### 3.7. Doradilla (Selaginella Lepidophylla (Hook. & Grev.) Spring)

*Selaginella lepidophylla (Hook. & Grev.) Spring* belongs to the *Selaginellaceae* family; its common names are ”doradilla” (goldenish), “siempre viva” (evergreen), “planta de la resurrección” (resurrection plant), and “flor de piedra” (stone flower). It is a Mexican native plant used in traditional medicine for gall and renal stones, diuresis, stomach, and liver inflammation [55,56]. Its effect on UL has been a reporter in the study in an animal model of UL, the results showed that the intervention with an extract of this plant (50 mg/kg) for 21 days decreased serum creatinine and oxalic acid, and improved glomerular filtration rate. In addition, these extracts also decreased the level of ROS in cortical tissue compared to the UL group [55]. Another study in the animal model showed that oral administration of an extract that contained mainly alkaloids of this plant (100 mg/kg) promoted an increase in the urinary sodium excretion [57].

This plant has been identified with some bioactive compounds such as flavonoids and alkaloids, which may be responsible for the diuretic effect in these extracts [57]. 

### 3.8. Diente de León (Taraxacum Officinale *L.*)

*Taraxacum officinale* L. is a wild plant belonging to the family *Compositae*; its common name is “Diente de León” (Dandelion). This plant has been used in traditional medicine as a treatment of several diseases due to its choleretic, diuretic, antiangiogenic, antirheumatic, antioxidants, and anti-inflammatory properties [58]. An in vitro study showed that an extract of this plant (1–8 mg/mL) promoted the inhibition of nucleation. Additionally, this study also reported that the effect of the use of taraxasterol (7.5 and 12.5 g/mL),a compound extract of this plant, caused significant decreases in CaOx compared to the UL group in a dose-dependent manner [59]. Other evidence in an animal model of UL induced by EG, showed that the administration of taraxasterol (2, 4, and 8 mg/kg) for 33 days decreased urine concentration of magnesium and oxalate. Moreover, this intervention increased urine pH and citrate concentration and prevented injury and inflammation in renal tissue. In addition, the increased activity of antioxidant enzymes such as SOD and GPx in serum compared to the UL group [60]. 

The main phytoconstituents of *Taraxacum officinale* L. extract are tocopherols, phenols, flavonoids, saponins, and sterols, major sesquiterpene lactones including taraxasterol, and trace elements (calcium, sodium, magnesium, and potassium) [58,59]. Thus, the presence of these compounds could be contributing to the antilithogenic effect through the disaggregation of mucoprotein suspension and crystallization promoters.

**Table 1 molecules-27-02008-t001:** Experimental evidence on plants used in Mexican traditional medicine used for prevention and treatment of urolithiasis.

Common Name	Binomial Nomenclature	Bioactive Compounds	Study Type	Study Design	Main Results	References
Chicalote amarillo	*Argemone mexicana* L.	3,4-dihidro-3-hidroxi-7- (7-metiloctil) naftalen-2 (1 H)	In vitro	A mix of reaction to induce nucleation or aggregation where it was incubated with methanol leaf extract (100 mg/mL)	Inhibited nucleation and aggregation compared to standard cystone drug	[34]
			In vivo	Wistar rats induced to UL by EG, and received following treatments for 13 days (*n* = 12 each group): Control group + water UL group+ water UL +cystone (750 mg/kg) UL+ methanolic extract (400 mg/kg) UL+ methanolic extract (750 mg/kg)	↓↓ Serum creatinine and calcium in extract groups ↑ Diuresis * in all groups	[35]
Palo amarillo	*Berberis trifoliata Hartw. ex Lindl.*	Berberine	In vitro	Nucleation of CaOx crystals was induced by a mix of calcium chloride and sodium oxalate, and incubated with different concentration of *B. trifoliata* methanolic extract (100–1000 µg/mL)	Dose-dependent inhibition of crystallization	[37]
			In vivo	Wistar rats induced to UL by zinc disc in bladder and received following treatments for 20 days (*n* = 6 each group): Control UL group UL + methanolic extract (50 mg/kg) UL + methanolic extract (100 mg/kg) UL + methanolic extract (150 mg/kg)	↓↓ Weight of the depositions around the implants with all doses	[38]
Planta de la insulina	*Costus mexicanus Liebm*	Lupeol and stigmasterol	In vitro	A mix to induce nucleation and growth COM crystals was generated and incubated with water or different concentrations of plant aqueous extract (0.15–1%)	↓ Mass of crystals and nucleation was delayed with a dose-dependent concentration	[41]
			In vivo	Wistar rats induced to UL by EG and received following treatments (*n* = 4, each group). Control: UL group + water (1 g/kg) UL group + Aqueous extract of stem of *C. igneus* (100 mg/kg) UL group + ethanolic extract of stem of *C. igneus* (100 mg/kg) UL group + Lupeol (50 mg/kg) UL group + Lupeol (100 mg/kg) UL group + Stigmasterol (50 mg/kg) UL group + Stigmasterol (100 mg/kg) UL + gallium nitrate (50 mg/kg)	↓ Serum levels of urea, uric acid, calcium, phosphate, and creatinine levels in lupeol and stigmasterol groups ↓ Urine levels of calcium, oxalate, creatinine, phosphate, and uric acid levels in all groups ↓ Calcium oxalate deposits in kidney in all groups	[42]
Cenizo	*Chenopodium album* L.	Flavonoids and saponins	In vitro	A mix to induce crystallization, nucleation and aggregation of CaOx crystals were generated and incubated with aqueous extract of the leaves (500 and 1000 μg/mL)	↓↓↓ Size of crystals Inhibited nucleation and aggregation Inhibited COM growth *	[43]
			In vivo	Adult Wistar rats model of UL induced by EG was used to administer CAME or CAAE from the leaves *Chenopodium album* (*n* = 6) during 28 days. Control group + water UL group + water UL group + Cystone (750 mg/kg) UL group + CAME (100 mg/kg) UL group + CAME (200 mg/kg) UL group + CAME (400 mg/kg) UL group + CAAE (100 mg/kg) UL group + CAAE (200 mg/kg) UL group + CAAE (400 mg/kg)	↑ Urine volume with CAME and CAAE (200 and 400 mg/kg) ↑ Urine pH with CAME and CAAE (400 mg/kg) ↓↓↓ Urinary levels of urea, uric acid, calcium, and phosphorus and creatinine with CAME (200 and 400 mg/kg) and CAAE (100, 200 and 400 mg/kg) ↓ Plasma levels of creatinine and urea with treatment with CAME and CAAE (400 mg/kg) ↓ Urine oxalates level with CAME and CAAE (400 mg/kg) ↓↓↓ Renal oxalate level with CAME and CAAE (400 mg/kg)	[44]
Visnaga	*Ammi visnaga* (L.) *Lam.*	Khellin and visnagin	In vitro	CaOx-supersaturated human urine was used to evaluate the effect of aqueous extract from the whole plant (200 μL and 600 μL) and from the seeds (200 μL and 600 μL) of AVL	Inhibition of the crystallization (extract of seeds at 200 µL and 600 µL) In the presence of extract a full AVL or seed there is a great increase in the CaOx dihydrate *	[47]
			In vitro	Cell lines (MDCK and LLC-PK1) were exposed to oxalate (300 µmol) and COM crystals (133 µg/cm^2^). Cells were incubated during 1, 3, 6, and 12 h as follows: Control + vehicle Control + Oxalate 300 µM Oxalate + KE (10, 50, 100, or 200 µg/mL) Control + COM COM + KE (10, 50, 100, or 200 µg/mL)	↓↓↓ Cellular damage (% LDH release) in LLC-PK1 cells (50, 100 and 200 µg/mL KE) ↓↓↓ Cellular damage (% LDH release) in MDCK cells (100 and 200 µg/mL KE)	[49]
			In vivo	Male Sprague-Dawley rats were induced to UL with EG. The animals were divided into the following experimental groups (*n* = 8 per group). The intervention included vehicle or KE. All treatments were administered orally for 14 days. Control group + vehicle UL group + vehicle UL group + KE (125 mg/kg) UL group + KE (250 mg/kg) UL group + KE (500 mg/kg)	↓↓ Deposition of CaOx crystal in kidneys ↑↑↑ Urinary excretion of citrate in all doses of KE ↓↓Urinary excretion of oxalate (KE 250 and 500 mg/kg) ↑ Urinary pH in all doses of KE ↑ Urinary volume and urinary calcium (KE 500 mg/kg)	[48]
Palo Azul	*Eysenhardtia polystachya (Ortega) Sarg.*	7-hydroxy-2′,4′,5′-trimethoxyisoflavone and 7-hydroxy-4′- methoxyisoflavone,	In vivo	Male Wistar rats induced UL by implantation of a zinc disc in bladder were divided into seven groups (*n* = 10, each group): Control group; Sham operated group and UL-induced group with different dosed of plant extract (25, 50, and 100 mg/kg)	↓↓↓ Stone deposition in all treatment groups, dose dependent ↑ Diuretic activity in all treatments groups, dose dependent	[50]
			In vivo	Female Wistar rats were divided into six treatments (*n* = 6). Control group received water (1 mL/kg) Furosemide-treated group (4 mg/kg) Groups treated with aqueous extract of *Eysenhardtia polystachya* (Ortega) Sarg at the doses of 125, 250, 500, and 750 mg/kg	↑ Urinary flow rate (500 and 750 mg/kg) ↑ Urinary excretion of sodium (750 mg/kg)	[51]
Doradilla	*Selaginella lepidophylla (Hook. & Grev.) Spring*	Flavonoids and alkaloids	In vivo	Wistar female rats were induced to UL by administration of EG. After this period, the rats were divided into two groups (*n* = 6, each group): UL and UL + CE from *Selaginella lepidophylla* (Hook. & Grev.) Spring (50 mg/kg).	↓↓↓ Urinary oxalic acid concentration compared to UL group ↑↑↑ Urinary flow rate ↑↑↑ Glomerular filtration rate in CE-treated compared with UL group ↑↑↑ Urinary excretions of sodium and potassium	[53]
			In vivo	Female healthy Wistar rats were divided into six groups (*n* = 6, each group): Control group (water 1 mL/kg), furosemide group (4 mg/kg), group treated with aqueous extract from *S. lepidophylla* (200 mg/kg), and groups treated with different concentrations of alkaloid fraction from *S. lepidophylla* (10, 40, and 100 mg/kg).	↑ Urinary excretion of sodium, potassium, and water in alkaloids fraction group	[55]
Diente de león	*Taraxacum officinale* L.	Tocopherols, phenols, flavonoids, saponins and sterols, including taraxasterol	In vitro	A model of CaOx crystallization in synthetic urine was generated by sodium oxalate. This was incubated with different concentrations of extract of *Taraxacum officinale* (1, 2, 4, and 8 mg/mL), taraxasterol (2.5, 5, 7.5, and 12.5 lg/mL), and PC (100, 150, 200, and 350 mg/mL)	↓ Nucleation of crystals with extract and taraxasterol and extract in all doses, dose dependent ↓↓↓ Number of CaOx crystals in a dose-dependent manner with extract and taraxasterol ↓↓↓ Diameter of CaOx crystals with extract and taraxasterol	[57]
			In vivo	A model of UL induced by ammonium chloride and EG in adult male Wistar albino rats. These were divided into the following groups (*n* = 6, each group): Control group, UL group, and UL with different concentrations of taraxasterol (2, 4, and 8 mg/kg), and UL with PC (2.5 g/kg).	Taraxasterol 2, 4, and 8 mg/kg and PC 2.5 g/kg ↓↓↓ Urinary oxalate levels. ↓↓ Score of inflammation in kidney Taraxasterol 2, 4, and 8 mg/kg ↓↓↓ Crystal deposits ↑↑ Urine pH ↑ Urinary citrate Taraxasterol 8 mg/kg: ↑↑ Superoxide dismutase and glutathione peroxidase in serum and kidney	[58]

Differences between groups are shown by *p* values: one arrow *p* < 0.05; two arrows *p* < 0.01; three arrows *p* < 0.001. * No *p*-value was reported. UL: urolithiasis; EG: ethylene glycol; CaOx: calcium oxalate; COM: calcium oxalate monohydrate; CAME: methanol extract; CAAE: aqueous extract of the leaves; AVL: *Ammi visnaga* L.; MDCK: Madin–Darby canine kidney collecting tubular epithelium cell line; LLC-PK1: porcine kidney proximal tubular epithelial cell line; KE: khella extract; CE: chloroform extract; PC: potassium citrate; LDH: lactate dehydrogenase. All doses are in mg/kg of body weight.

## 4. Molecular Mechanism of Bioactive Compounds Present in Medicinal Plants

The beneficial effect of the use of these medicinal plants has been documented in several ancient texts. However, few studies have demonstrated their traditional medicinal effects. Thus, this lack of information is a limitation of the knowledge of the mechanism of action of these medicinal plants. The presence of several bioactive compounds in these plants may be responsible for their beneficial effects [34,61]. A few of the main compounds contained in their chemical composition are flavonoids, phytosterols, saponins, furanochromones, alkaloids, and terpenoids. These types of compounds can have a positive effect due to their antioxidant and anti-inflammatory activity, as well as antimicrobial, litholytic, antispasmodic, and diuretic properties (Figure 2). Thus, these properties could be related to the reduction in urinary saturation by inhibiting crystallization, nucleation, and crystal aggregation [34,62].

### 4.1. Flavonoids

Flavonoids are a group of phenolic compounds with low molecular weight polyphenolic structures. Their main classes are flavonols, flavones, flavanones, anthocyanins, isoflavones, and flavanols. The flavonoids that promote beneficial effects on UL are rutin, catechin, epicatechin, and diosmin [63,64]. Flavonoids promote diuretic, antioxidant, anti-apoptotic, anti-infection, anti-inflammatory, and antibacterial effects. Flavonoids promote an inhibitory effect on urinary stone formation by their anti-inflammatory and antioxidant effects, which interfere with the process of epithelial cell damage induced by CaOx crystals and exert inhibitory effects on the inflammation. Some in vitro and in vivo studies suggested that the antioxidant activity of flavonoids played an important role in the prevention of the accumulation of CaOx by the reduction in ROS levels, which promoted protection against oxidative renal damage [64,65]. Additionally, it has been reported that the use of flavonoids inhibited CaOx crystal deposition in the urine of animal models [66,67,68].

The antioxidant activities of flavonoids occur because these compounds are capable of scavenging ROS and can induce the gene expression of antioxidant enzymes and chelation of transition metals, which decrease membrane lipid peroxidation and renal cell injury [64]. Flavonoids can also inhibit the enzyme activity of the angiotensin-converting enzyme (ACE), which forms part of the renin–angiotensin–aldosterone system. The inhibition is due to the flavonoids that can chelate minerals such as zinc, which is present within the active center of ACE. Thus, the inhibition promotes beneficial effects; furthermore, it significantly reduces CaOx crystal deposition and renal inflammation, and in turn, decreases the ROS levels [64]. 

Additionally, flavonoids have shown anti-inflammatory activity by suppressing NF-κB activation, which in turn reduces the production of prostaglandines-E2, cytokines IL-1β, IL-6, and TNF-α, and also inhibits cyclooxygenases enzymes. Moreover, they have effects inhibiting the synthesis of oxalate and increasing the bioavailability of nitric oxide to chelation of calcium through the cGMP (3′, 5′ cyclic guanosine monophosphate) pathway. For example, quercetin presented anti-inflammatory activity by the inhibition of NF-κB, which in turn is related to the important mechanisms of diuretic activity in lithogenesis [64,65].

### 4.2. Phytosterols

Phytosterols are plant compounds with similar structures to cholesterol. Among these compounds are β-sitosterol, lupeol, campesterol, stigmasterol, stigmastanol, and cycloartenol. Several phytosterol-rich plants have been shown to possess a wide range of therapeutic activities [69]. The administration of isolated lupeol and stigmasterol in a study in an animal model of hyperoxaluria showed a significant decrease in the levels of urine and serum calcium, oxalate, phosphate, uric acid, creatinine level, and also CaOx crystal formation in the kidney [44]. These beneficial effects may be attributed to various effects to phytosterols such as anti-inflammatory and antioxidant actions; for example, β-sitosterol increased the activities of the antioxidant enzymes SOD and GPx under OS. The antioxidant activity during UL plays an important role in the oxidative damage in the formation of urinary stones. In addition, the inflammatory effect of β-sitosterol and campesterol promoted a significant reduction in prostaglandin E and prostaglandins, which also protect the tissue injury [69]. 

### 4.3. Saponins

The saponins are mostly triterpene glycosides with triterpenes or steroids in aglycone. These types of compounds are present in a great number of medicinal herbs with antilithiatic and diuretic properties. Studies reported that saponins play an important role in preventing the formation of urinary stones [70,71]. The anti-crystallization property of saponins is due to their ability to act to inhibit crystal–cell interaction, which in turn reduces crystal retention in the urinary tract [72]. Moreover, it has been shown that this effect could be due to saponins showing a higher rate of ionization and Ca^2+^ chelation, thereby increasing the solubility of CaOx crystals and finally decreasing the formation of urinary stones [73].

### 4.4. γ-Pyrones (Furanochromone Derivatives)

The pyranocoumarins represented by an angular-type dihydro-pyranocoumarin glucoside include khellin and visnagin [47,50,51]. These types of compounds have been shown to protect the renal epithelial cell damage from oxalate and calcium crystals. Furthermore, they also prevent the oxalate formation that is associated with hyperoxaluria by increasing the urinary pH. These beneficial effects may be related to the anti-inflammatory effect of visnagin. This compound inhibits transcription factors such as NF-κB, which in turn decreases gene expression and releases pro-inflammatory molecules, including TNF-α, IL-1β, and IFNγ [47].

### 4.5. Alkaloids

Alkaloids are organic compounds that contain a heterocyclic nitrogen ring. These compounds have been widely studied by their pharmacological properties. One of these compounds that showed beneficial effects is berberine. Evidence suggests that it can increase the urine volume and the Na^+^ and K^+^ excretion similar to the standard diuretic drug (hydrochlorothiazide). In addition, it has been shown that treatment with berberine decreases urine Ca^2+^ content. This has great relevance in urinary supersaturation due to the excess of calcium being the primary requisition for crystal precipitation and one of the major risk factors for stone development [41]. Thus, berberine may prevent the formation of urinary stones by increasing the excretion of minerals in small particles [40]. 

### 4.6. Terpenoids

Terpenoids are compounds distributed in natural products such as plants. Some evidence has shown their important pharmacological properties. Some compounds classified as terpenoids that can act against UL are cafestol, kahweol, and taraxasterol. Many studies have shown the antilithogenic activity of these compounds; their effect is related to the increase in urinary citrate. Citrate is a compound that can inhibit crystal deposition and growth in renal tissue [74,75,76]. Additionally, these types of compounds stimulate the relaxation of the smooth muscle of the urothelium, which promotes a decrease in their size and facilitates expulsion of the urinary stones [77]. 

Terpenoids also show antioxidant and anti-inflammatory activities; there is evidence that antioxidant activity in cell and animal models involves triggering the upregulation of key antioxidant enzymes [76]. Thus, they can decrease ROS levels by promoting the expression of endogenous antioxidants such as SOD and GPx. In addition, an inflammatory effect can inhibit the production of ROS and the pro-inflammatory molecules including TNF-α, IL-6, and IL-1β. There is evidence about the potential anti-inflammatory effect of these compounds, which is associated with a decrease in tubular damage and acute necrosis in the histopathological analysis of renal tissue [74].

## 5. Materials and Methods

### Literature Search

Electronic databases, including PubMed, Science Direct, and Scopus, were searched for dietary plants and their bioactive compounds used for the prevention and management of urolithiasis from 2005 to December 2017. The grey literature was searched using the website Google Scholar. The terms used were “urolithiasis”, “kidney stone”, “traditional medicine”, “phytotherapy”, “Mexico”, and “Mexican”. Additionally, Spanish words such as “litiasis urinaria”, “urolitiasis”, “medicina traditional”, or “plantas medicinales” were also used. The names of plants were authenticated using the Plant List and Royal Botanical Garden, Kew databases. Databases of the Autonomous University of Yucatán, Yucatan Scientific Research Center (CICY), and Center for Research and Advanced Studies (CINVESTAV) of the National Polytechnic Institute were mainly considered for collecting information about medicinal plants. The retrieved articles were subclassified into in vitro and in vivo studies. The studies included were evaluated with respect to the potential of the plant to be used as a dietary agent, the phytochemical composition of the plant, the kind of kidney stone that the dietary agent is effective on, as well as underlying mechanisms of action.

## 6. Conclusions

The use of natural products plays a key role in the prevention and treatment of various diseases. Among those are different types of plants used in traditional medicine that promote beneficial effects, which could be attributed to the quantity and quality of bioactive compounds present in their chemical composition. In Mexico, there is a large use of medicinal plants for many pathologies, including UL. However, so far, the relationship between their use and the prevention of UL has not been concretely established. Thus, this review presents the scientific evidence about these medicinal plants used in Mexico and their bioactive compounds and possible mechanisms for the prevention and treatment of UL in preclinical studies. This review included in vitro and in vivo studies using plants such as *Argemone mexicana* L., *Berberis trifoliata Hartw. ex Lindl.*, *Costus mexicanus Liebm.*, *Chenopodium album* L., *Ammi visnaga* (L.) *Lam.*, *Taraxacum officinale* L. On the other hand, other plants such as *Eysenhardtia polystachya (Ortega) Sarg and Selaginella lepidophylla (Hook. & Grev.) Spring* were evaluated in some in vivo studies. The main bioactive compounds present in these medicinal plants were 3,4-dihydro-3-dihydro-3-hydroxy-7-(7-methylocetyl) naphthalen-2 (1H), berberine, lupeol and stigmasterol, flavonoids and saponins, khellin and visnagin, 7-hydroxy-2′,4′,5′-trimethoxyisoflavone and 7-hydroxy-4′-methoxyisoflavone, alkaloids, tocopherols, phenols, and sterols, including taraxasterol. The molecular mechanisms of these bioactive compounds are related to antioxidant, anti-inflammatory, and antimicrobial properties.

In conclusion, this review reveals the results obtained from the available literature about the medicinal plants used in Mexico and the possible molecular mechanisms of action due to their content of bioactive compounds. These studies seem promising for the prevention and intervention of UL, although only preclinical studies were identified. However, clinical trial investigations are needed to confirm both the efficacy and safety of the use of medicinal plants for the prevention and treatment of UL.

## Figures and Tables

**Figure 1 molecules-27-02008-f001:**
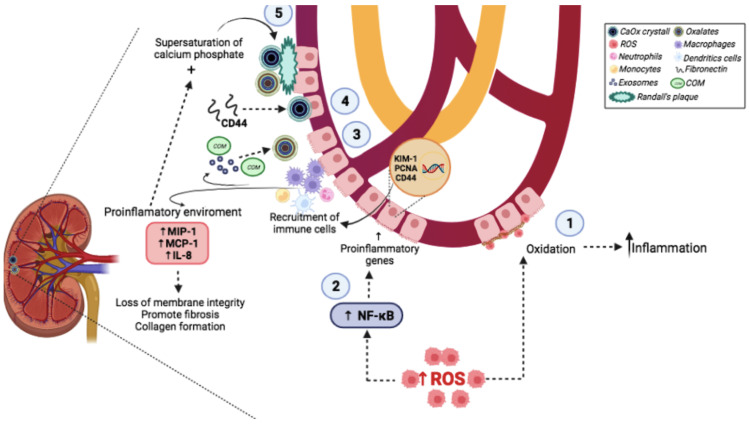
Molecular mechanisms involved in the formation of urinary stones. (1) Excessive ROS levels promote oxidation of renal cell membrane components, and this causes biochemical alterations. (2) Overexpression of pro-inflammatory genes through NF-κB by ROS. (3) Injury in kidney by chemokines and immune cells such as monocytes, macrophages, and neutrophils. (4) Macrophages in development of CaOx crystals. (5) Pro-inflammatory state and oxidative damage induces Randall’s plaque formation by progressive supersaturation of calcium phosphate induction. ROS: reactive oxygen species; NF-κB: nuclear transcription factor κB; CaOx: calcium oxalates; COM: calcium oxalate monohydrate; MIP-1: macrophage inhibitory protein1; MCP-1, monocyte chemoattractant protein1; interleukin8; KIM-1: kidney injury molecule 1; PNCA: Proliferating cell nuclear antigen; CD44: Homing cell adhesion molecule.

**Figure 2 molecules-27-02008-f002:**
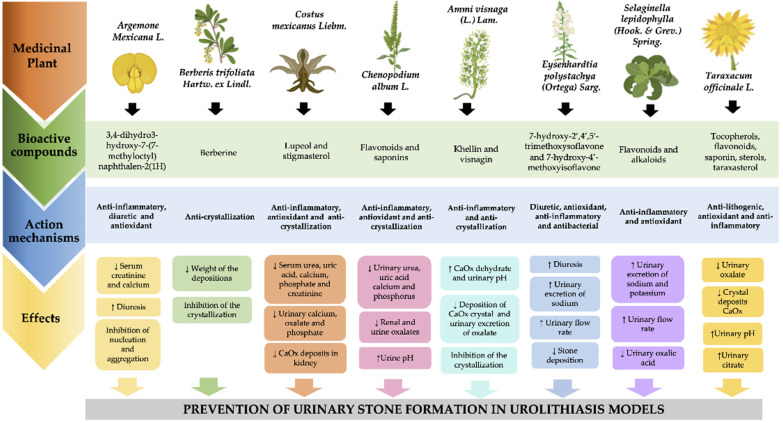
Molecular mechanisms involved in beneficial effects of medicinal plants through their bioactive compounds. CaOx: Oxalate calcium.

## Data Availability

Not applicable.

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
