# Peer review of "Plants Used in Mexican Traditional Medicine for the Management of Urolithiasis: A Review of Preclinical Evidence, Bioactive Compounds, and Molecular Mechanisms"

_molecules, 2022, doi:10.3390/molecules27062008_

Round 1
Reviewer 1 Report
The manuscript “Plants used in Mexican traditional medicine for the management of urolithiasis: A review of preclinical evidence, bioactive compounds and molecular mechanism” describes the available reports on the effects of some Mexican plants in treatment of urolithiasis, a disease associated with the formation of stones in urinary tract. The plants are rich in containing various bioactives and phenolics which are supportive to heals various dis-functionalities including urolithiasis. The manuscript is well written and the information gathered is scientifically important. However, there is some more information need to be added to improve the article.
- 148-151, the scientific names Argemone mexicana L., Berberis trifoliata, Costus mexicanus Liebm, Henopodium album, Ammi visnaga, Eysenhardtia polystachya, Selaginella lepidophylla, Taraxacum officinale should be italic.
- There are some minor grammatical errors, e.g., Line 164, Argemone mexicana should be italic.
- Could the authors add information table containing the traditional uptake process/ formula of consuming the plants by people? This table also may include the doses of the plants consumption for the problems.
- In Figure 2. Molecular mechanism involving in beneficial effects of medicinal plants through its bioactive compounds, the authors are suggested to add the detail mechanisms of the compounds in reducing urolithiasis. It will be the key finding of the article as the general functions are already discussed in many previous reports.
Author Response
Reviewer 1
The manuscript “Plants used in Mexican traditional medicine for the management of urolithiasis: A review of preclinical evidence, bioactive compounds and molecular mechanism” describes the available reports on the effects of some Mexican plants in treatment of urolithiasis, a disease associated with the formation of stones in urinary tract. The plants are rich in containing various bioactives and phenolics which are supportive to heals various dis-functionalities including urolithiasis. The manuscript is well written and the information gathered is scientifically important. However, there is some more information need to be added to improve the article.
Q1. 148-151, the scientific names Argemone mexicana L., Berberis trifoliata, Costus mexicanus Liebm, Chenopodium album, Ammi visnaga, Eysenhardtia polystachya, Selaginella lepidophylla, Taraxacum officinale should be italic.
R1. Thank you for your comment. We modified the scientific names in the italic form (now lines 158-161).
Q2. There are some minor grammatical errors, e.g., Line 164, Argemone mexicana should be italic.
R2. Thank you for your comment. We modified the scientific names in the italic form in all cases.
Q3. Could the authors add information table containing the traditional uptake process/ formula of consuming the plants by people? This table also may include the doses of the plants consumption for the problems.
R3. As you suggested we added a table (Supplementary table 1) with traditional uptake process and the doses.
Q4. In Figure 2. Molecular mechanism involving in beneficial effects of medicinal plants through its bioactive compounds, the authors are suggested to add the detail mechanisms of the compounds in reducing urolithiasis. It will be the key finding of the article as the general functions are already discussed in many previous reports.
R4. Thank you for your comment as you suggested in figure 2 we added information about the molecular mechanism involving in the beneficial effects of each plant.
Reviewer 2 Report
The manuscript titled “Plants used in Mexican traditional medicine for the management of urolithiasis: A review of preclinical evidence, bioactive compounds and molecular mechanism” summarizes recent information on the bioactive potential of Mexican plants and their role in treating urolithiasis. The topic is of high interest, and the manuscript is well-written in general. However, as suggested below, some points should be considered in terms of structure, language, and information. Therefore, I recommend major revisions.
General comment: Grammar needs improvement.
General comment: The section numbering is really confusing, as the authors jump from 1 to 4. I believe 1 should be a brief introduction, and from 2 onwards they should continue with the topics.
General comment: in vivo and in vitro should always be written in italic. This needs to be fixed throughout the whole manuscript.
Line 97 – It should be “promotes.”
Lines 99-100 – The authors mention “some studies,” but cite only 1 reference…
Line 134 – It should be “pharmacological.”
Line 152: Which macromolecules?
Lines 156-157: “recognition of the principle between the various molecules present in each plant where each could have individual effects.” This sentence is not very clear. Are the authors referring to the structure/activity relationship of the bioactives present in these plants?
Lines 158-162: I believe this should be moved once the authors create a single introduction section.
Line 179: phenolics and flavonoids are mentioned as if they were completely different things. Flavonoids are a sub-class of phenolic compounds. Therefore, the way this is written does not make sense.
Line 220: It should be “an in vitro study.” Same in other places.
Lines 228-229: The authors should be more specific when explaining why these compounds are associated with such effects. Only having them in the extracts does not guarantee this range of effects.
Line 300: It should be “traditional medicinal effects.”
Lines 304-305: not all those motioned are phenolic compounds. This needs to be checked.
Line 320: played an important what?
Author Response
Reviewer 2.
The manuscript titled “Plants used in Mexican traditional medicine for the management of urolithiasis: A review of preclinical evidence, bioactive compounds and molecular mechanism” summarizes recent information on the bioactive potential of Mexican plants and their role in treating urolithiasis. The topic is of high interest, and the manuscript is well-written in general. However, as suggested below, some points should be considered in terms of structure, language, and information. Therefore, I recommend major revisions.
Q1. General comment: Grammar needs improvement.
R1. Thank you for your comment; as you suggested we rechecked the grammar of the manuscript.
Q2. General comment: The section numbering is really confusing, as the authors jump from 1 to 4. I believe 1 should be a brief introduction, and from 2 onwards they should continue with the topics.
R2. Thank you for observation. We modified the introduction section (now lines 41-99), and changed the numeration of all sections.
Q3. General comment: in vivo and in vitro should always be written in italic. This needs to be fixed throughout the whole manuscript.
R3. Thank you for comment. We have modified these terms to italic form in all cases.
Q4. Line 97 – It should be “promotes.”
R4. We have changed to “promotes” in this line (now line 121).
Q5. Lines 99-100 – The authors mention “some studies,” but cite only 1 reference…
R5. Thank you for observation, we added the references about this information (now lines 123-125).
Q6. Line 134 – It should be “pharmacological.”
R6. Thank you for comment, we changed to “pharmacological” (now line 79).
Q7. Line 152: Which macromolecules?
R7. We modified and added the information in this line (now lines 161-163).
Q8. Lines 156-157: “recognition of the principle between the various molecules present in each plant where each could have individual effects.” This sentence is not very clear. Are the authors referring to the structure/activity relationship of the bioactives present in these plants?
R8. We modified the sentence to be clear (now lines 93-94).
Q9. Lines 158-162: I believe this should be moved once the authors create a single introduction section.
R9. As you suggested we modified the introduction section (now lines 41-99) and move the information to the end of this section (now lines 95-99).
Q10. Line 179: phenolics and flavonoids are mentioned as if they were completely different things. Flavonoids are a sub-class of phenolic compounds. Therefore, the way this is written does not make sense.
R10. We totally agree with your comment, we modified the sentence (now line 183).
Q11. Line 220: It should be “an in vitro study.” Same in other places.
R11. Thank you for your comment, as you suggested we modified this sentence in all cases of the manuscript.
Q12. Lines 228-229: The authors should be more specific when explaining why these compounds are associated with such effects. Only having them in the extracts does not guarantee this range of effects.
R12. We added information about the effects caused by these types of compounds to decreased urinary stones (now lines 235-237).
Q13. Line 300: It should be “traditional medicinal effects.”
R13. As you suggested, we modified the phrase (now line 309-310).
Q14. Lines 304-305: not all those motioned are phenolic compounds. This needs to be checked.
R14. We totally agree with your comment, we modified the text to be clear (now line 313).
Q15. Line 320: played an important what?
R15. Thank you for your observation, we added the word “role” (now line 328).
Round 2
Reviewer 1 Report
The authors have improved the manuscript as per reviewer suggestions and now could be accepted.
Reviewer 2 Report
The authors have included all suggestions and significantly improved their manuscript. Therefore, I recommend publication in the present form.